# Removal of cyanobacteria from a water supply reservoir by sedimentation using flocculants and suspended solids as ballast: Case of Legedadi Reservoir (Ethiopia)

Hanna Habtemariam[1]*, Demeke Kifle[2], Seyoum Leta[1], Maíra Mucci[3], Miquel Lürling[3,4]

**1** Center for Environmental Science, Addis Ababa University, Addis Ababa, Ethiopia, **2** Department of Zoological Science, Addis Ababa University, Addis Ababa, Ethiopia, **3** Department of Environmental Sciences, Aquatic Ecology & Water Quality Management Group, Wageningen University, Wageningen, The Netherlands, **4** Department of Aquatic Ecology, Netherlands Institute of Ecology (NIOO-KNAW), Wageningen, The Netherlands

* hanhabtemariam@gmail.com

## Abstract

The massive growth of potentially toxic cyanobacteria in water supply reservoirs, such as Legedadi Reservoir (Ethiopia), poses a huge burden to water purification units and represents a serious threat to public health. In this study, we evaluated the efficiency of the flocculants/coagulants chitosan, *Moringa oleifera* seed (MOS), and poly-aluminium chloride (PAC) in settling cyanobacterial species present in the Legedadi Reservoir. We also tested whether coagulant-treated reservoir water promotes cyanobacteria growth. Our data showed that suspended solids in the turbid reservoir acted as ballast, thereby enhancing settling and hence the removal of cyanobacterial species coagulated with chitosan, *Moringa oleifera* seed, or their combination. Compared to other coagulants, MOS of 30 mg/L concentration, with the removal efficiency of 93.6%, was the most effective in removing cyanobacterial species without causing cell lysis. Contrary to our expectation, PAC was the least effective coagulant. Moreover, reservoir water treated with MOS alone or MOS combined with chitosan did not support any growth of cyanobacteria during the first two weeks of the experiment. Our data indicate that the efficacy of a flocculant/coagulant in the removal of cyanobacteria is influenced by the uniqueness of individual lakes/reservoirs, implying that mitigation methods should consider the unique characteristic of the lake/reservoir.

## 1. Introduction

Eutrophication-related deterioration of the quality of freshwaters has become an environmental issue of global concern [1] and will remain to be the most important water quality problem in the future [2]. Global climate change and anthropogenic nutrient input have resulted in frequent occurrences of cyanobacterial blooms [3,4]. Cyanobacterial bloom is a serious concern since it results in changes in the odor and taste of water. It is associated with the production of

**Data Availability Statement:** All relevant data are within the manuscript and its Supporting Information files.

**Funding:** This research did not receive any specific grant from funding agencies in the public, commercial, or not for profit sectors.

**Competing interests:** The authors have declared that no competing interests exist.

potent toxins, which present a threat to the health of humans and animals [5]. Consequently, cyanobacterial blooms may impair recreational activities, such as angling, boating, swimming, irrigation, aquaculture, and drinking water production [6].

Sixty percent of the population of Addis Ababa city, which is 2.75 Million people, depends on the Legedadi and Geffersa Reservoirs as sources of drinking water supply. However, potentially toxic and persistent cyanobacterial blooms dominated by *Microcystis aeruginosa* and several *Anabaena* spp. have recurred annually in Legedadi Reservoir [7,8]. As cyanobacterial blooms in eutrophic water bodies present a high public health risk, the immediate reduction of the potential hazard is imperative. Even though the reservoir has suffered from eutrophication-related water quality problems for decades, reports on efforts made to control the cyanobacteria using environmentally friendly restoration techniques are absent. However, the application of algaecides like copper sulfate has been practiced in the study reservoir for more than three decades. The main drawback of using copper sulfate is the lysis of cyanobacterial cells, which results in the release of toxins into the water bodies. Cell lysis caused by algaecides may lead to intoxication of aquatic organisms and eventually of humans [9].

To tackle cyanobacteria-related problems and restore water bodies, the first rational option is the reduction of external nutrient loading. However, this option is not always feasible [10], especially in developing countries like Ethiopia, where agriculture is expanding and the application of fertilizers is widely practiced. The disposal of untreated wastewater into reservoirs/ lakes and the huge cost incurred by the establishment of proper wastewater treatment plants (WWTP) complicate the problem further [11,12]. Therefore, effect-oriented measures would be suitable for controlling nuisance/harmful cyanobacteria, especially for developing countries where control of external nutrient loading is not economically a preferred option at this moment. An effect-oriented technique has its own drawbacks, including its short-lived effects and the need to repeat it regularly. Thus, there is a need to look for fast, easy, cheap, and safe technologies as curative/effect-oriented methods. To this end geochemical engineering approaches, such as "floc and lock" and "floc and sink", have been developed [13,14]. In the floc and sink technique, flocculants/coagulants such as chitosan, poly aluminium chloride (PAC), iron chloride, and *Moringa oleifera* seed-extract (MOS) with ballast materials (clays, natural soils, lanthanum-modified bentonite, and Phoslock) have been used to effectively coagulate cyanobacteria as intact cells and to settle to the sediment of different lakes/reservoirs [14].

The 'floc and lock' technique has been applied effectively in two isolated and stratifying lakes, Lake Rauwbraken [13,15] and Lake De Kuil [16] in the Netherlands, while 'floc and sink' has been tested effectively in an isolated bay, Lake Taihu [17] in China.

Prior to implementing such measures, in-depth research and recognition of the unique feature of the water body of concern is needed [18,19]. Those include hydro-morphological and physico-chemical features, the biology of the system, and for instance availability of local soils with natural P binding capacities [20]. In Legedadi Reservoir, the high turbidity caused by particulate materials entering the reservoir through rivers was considered a unique feature. In previous studies, ballast materials such as clay and soil were added with a coagulant to lake/ reservoir water [14,20]. The high turbidity of the Legedadi Reservoir, however, was expected to bring sufficient autochthonous ballast such that, adding additional ballast is not needed. Accordingly, the floc and sink technique was implemented by directly adding only a flocculant/coagulant to the Legedadi Reservoir water.

The addition of a coagulant to reservoir water was expected to clear the water column from cyanobacteria and suspended solids. Consequently, the reservoir water was expected to become clear and light limitation would no longer prevail. As Legedadi Reservoir is a nutrient-rich reservoir, a newly cleared water column condition could allow light penetration to greater

depth thereby promoting cyanobacteria growth. Therefore, an experiment was conducted to determine what could possibly happen in terms of cyanobacterial growth as the reservoir water becomes clear.

We hypothesized that PAC, MOS extract, and chitosan in a turbid water body would effectively remove cyanobacteria without the need for adding external ballast material. It was also hypothesized that, the reservoir water treated with MOS extract would lead to delayed growth of cyanobacteria due to the antibacterial effect of MOS. The laboratory experiments were conducted using reservoir samples to evaluate the efficacy of different doses of PAC, MOS extract, chitosan, and a blend of chitosan and MOS extract for settling and removing cyanobacterial biomass. Furthermore, we also examined whether or not the effectively treated reservoir water was capable of supporting the growth of cyanobacteria. The growth experiment was expected to give an insight into the longevity of the intervention method.

## 2. Materials and methods

### 2.1. Study area

Legedadi Reservoir is one of the major drinking water sources for the capital city of Ethiopia, Addis Ababa. It is administered by Addis Ababa Water and Sewerage Authority [21]. The reservoir is located at an altitude of 2450 m a.s.l. and at a geographical position of $9^0$ 01' - $9^0$ 13' N latitude and $38^0$ 60' - $39^0$ 07' E longitude (Fig 1). The reservoir, was constructed in 1967, with an initial storage capacity of 45.9 $Mm^3$ and a surface area of 5.324 $km^2$ [21]. It has a total catchment area of 205.7 $km^2$. It has a mean and maximum water depth of 4 m and 30 m,

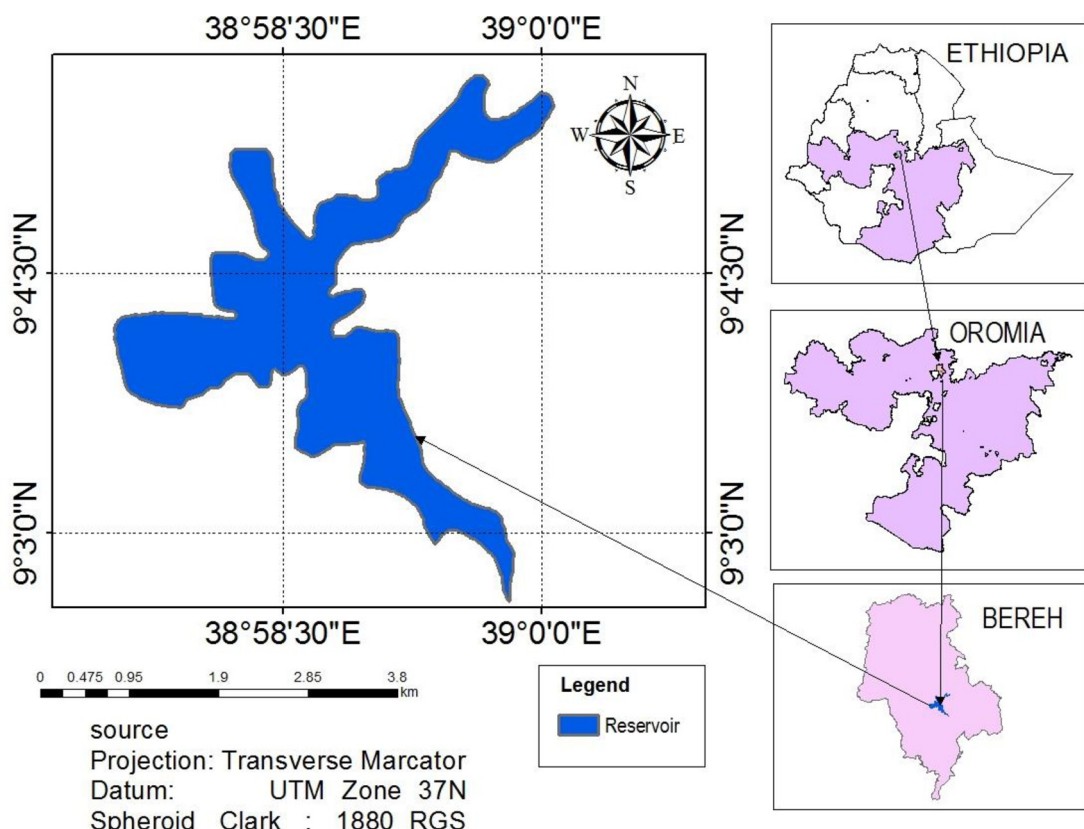

**Fig 1. Map showing the location of the Legedadi Reservoir.**

respectively. The reservoir has an outflow rate to the water treatment plant of 126,666 m$^3$/day and a water retention time of 325 days [21].

## 2.2. Reservoir water, chemicals and other materials

The water sample used for the experiment was collected around the dam in January, 2018. During sampling month, the mean concentration (μg/L) of total phosphorous (TP) and soluble reactive phosphate-P (SRP) were 597 and 167 respectively. While the mean concentration (μg/L) of nitrate-N and ammonium-N were 276 and 3.06 respectively. In the reservoir, the mean concentration (mg/L) of total suspended solids (TSS) was 390 and the mean concentration (mg/L) of total dissolved solids (TDS) was 77. The mean value of pH and conductivity (K$_{25}$, μS cm$^{-1}$) were 7.9 and 29.9 respectively. Cyanobacteria constituted 98% of total phytoplankton taxa, with species of *Dolichospermum* and *Microcystis* dominating the phytoplankton assemblages [22].

Dry *Moringa oleifera* seeds (MOS) with the pod, a natural coagulant, were purchased from Addis Ababa (Ethiopia). According to [23], MOS obtained from Ethiopia, on a dry matter basis of (g/100g) had the chemical composition of 6.1 ± 0.2, moisture content, 41.4 ± 1.6, oil content, 42.6 ± 1.4, protein content, 5.1 ± 0.3 crude fiber content, and 55.6 ± 1.5 crude protein content.

The healthy seeds (about 1.0 cm long) were selected after they were de-shelled and oven-dried at 105˚C for 2 hrs. The kernels were crushed in a mortar and sieved through 300 μm to produce a powder with particles of ∼300 μm diameter [24]. The powder was stored in an air-tight container and used within a month. To extract the active cationic coagulating proteins, 5 g of the seed powder was suspended in 100 mL of 1.0 mol/L NaCl solution and stirred using a magnetic stirrer for 30 min. The solution (MOS modifier) was filtered through a membrane microfiber filter paper of 0.45 μm pore size, yielding a stock of 2.8 ± 0.032 g/L of MOS extract. An additional stock of 100 mL of 1.0 mol/L NaCl Milli-Q water was prepared as a control [24,25].

The coagulant Chitosan was obtained from Polymar Ciência e Nutrição S/A (Ceara', Brazil). First, in 200 mL flasks, 100 mg chitosan was dissolved in 20 mL Milli-Q water. Then, the solution was acidified by adding 100 μL of 96% acetic acid (Merck analytical grade). The solution was diluted to 100 mL with Milli-Q water, which yielded a stock of 1 g/L chitosan modifier. An additional stock of 100 μL of 96% acetic acid was added to 100 mL Milli-Q water and used as control [26]. The coagulant poly aluminium chloride (PAC, Al$_n$(OH)$_m$Cl$_{3n-m}$, density 1.37 kg/L, 8.9% Al, 21.0% Cl) was provided by Dr G. Waajen (Regional Water Authority Brabantse Delta).

## 2.3. Floc and sink assays

The 'floc and sink' assay was performed using different doses of coagulants/flocculants to coagulate and settle cyanobacterial species found in samples from Legedadi Reservoir. The reservoir has a mean annual turbidity (NTU) of 433 and exhibited a year-round occurrence and dominance of *Microcystis aeruginosa* and *Anabaena* (currently known as *Dolichospermum*) spp.

Initially, to maintain the growth of cyanobacteria, WC medium with vitamins added (H, biotin, and B12, cyanocobalamin, at 50 ng/L and B1, thiamine HCL, at 100 ng/L) [27], was added to the reservoir samples.

At the start of the experiment, chlorophyll-a (Chl-a) of 130 μg/L was recorded and cyanobacteria were in good condition as reflected by their Photosystem II efficiency (PSII) of 0.66 (± 0.01).The initial Chl-a concentration (μg/L) and PSII efficiency were determined using a phytoplankton analyzer (PHYTO-PAM, Heinz Walz GmbH, Effeltrich, Germany).

All experiments were conducted as follows: 11 mL Legedadi Reservoir sample was transferred into 15 mL glass tubes. Samples were treated with the coagulants/flocculants (PAC, MOs, chitosan, an effective dose combination of chitosan and MOS) or left untreated (controls). After dosing, the contents in each test tube were mixed briefly using a glass rod. Then, the samples were placed on a laboratory bench at room temperature and under still conditions [26].

After 1 h, 2 mL samples were taken from the top and bottom of the tubes using a 10 mL Eppendorf reference pipet. The 2 mL sample was used for the determination of Chl-a concentration and PSII efficiency as a measure of the health of the cyanobacterial cells. Then, the 2 mL sample was diluted 10 times with Milli-Q water and turbidity was measured using a turbidity meter (Hach 2100 P). The pH was measured in the glass tubes using a WTW pH meter.

PAC was dosed at 0, 1, 2, 4, and 8 mg Al/L and Chitosan was dosed at 0, 1, 2, 4, and 8 mg/L, while MOS extract was first dosed at 0, 10, 50, 100, 140 mg/L, and then the dosing was narrowed to a range of 0–50 (0, 10, 20, 30, 40, & 50) mg/L. Immediately after dosing, the contents in each test tube were mixed briefly using a glass rod. In all the four series of experiments, controls, and treatments were run in triplicate. Tubes were left untouched for 1 h, where after 2 mL top and 2 mL bottom samples were taken and analyzed as outlined above.

## 2.4 Treated reservoir water and its effect on cyanobacterial re-growth

In the second experiment, the effect of treated reservoir water (after treatment with an effective dose of MOS (30 mg/L) on cyanobacteria growth was assessed. This experiment aimed to test the hypothesis that the treatment lowered turbidity of the reservoir water (RW) to such an extent that light would not be limiting to cyanobacteria. To this end, a 26 days growth experiment was conducted using three cyanobacterial cultures: *Microcystis aeruginosa* PCC 7820, obtained from Pasteur Culture Collection (Institute Pasteur, Paris, France); *Anabaena flos-aquae* SAG 30.87 obtained from the Culture Collection of Algae at Gottingen University (Germany) and a *Microcystis* sp. culture (*Microcystis* sp. which was collected and isolated from Legedadi Reservoir, Ethiopia).

The three cyanobacteria were cultured in triplicate using three media, namely, modified WC medium [26], unaltered (raw) reservoir water (RRW) and treated reservoir water (TRW). In total, 27 replicates were used in this experiment. The experiment was conducted in 100 mL sterile Erlenmeyer flasks to which either 50 mL of TRW, 50 mL RRW, or 50 mL of autoclaved and cooled WC medium was added. Flasks with TRW or RRW received nutrients similar to those in the WC medium.

Then an inoculum of cyanobacterial cultures with Chl-a concentration of 20 (± 0.1) μg/L was added to the designated flask. One flasks with 50 mL RRW without addition of inoculum was included as additional control. Flasks were closed with a cellulose plug and placed at random in a Gallenkamp ORBI-SAFE Netwise Orbital Incubator at 25°C, 40 rpm shaking and in a 12:12 h light: dark cycle. The light: dark cycle was programmed in such a way that light intensity increased gradually to a maximum of 130 μmol quanta m$^{-2}$ s$^{-1}$ and subsequently decreased again to darkness, which resulted in daily average light intensity of ∼65 μmol quanta m$^{-2}$ s$^{-1}$. The light intensity within the incubator was measured at 22 locations and the experimental flask was swirled every time before sampling.

Samples were taken from the cultures initially and after 1, 2, 3, 6, 8, 11, 14, 19, and 26 days and were analyzed for Chl-a concentration and PSII efficiency using the PhytoPAM.

## 2.5 Data analysis

To examine the differences in the efficacy of coagulants/flocculants, one-way ANOVA at 95% confidence level was used. Significant differences were determined using a Tukey post hoc

comparison test ($p < 0.05$). As data on PSII efficiency failed normality test, Kruskal-Wallis One Way Analysis of Variance on Ranks was run instead. As growth rate also failed the normality test, Friedman Repeated Measures Analysis of Variance on Ranks was made to determine whether there was a significant difference among cyanobacteria and growth media. Significant differences were determined using a Tukey post hoc comparison test ($p < 0.05$).

The Chl-a data over time were analyzed per replicate by fitting the logistic growth model in which the rate of 'population increase' ($dA/dt$) is a function of the population density determined by Chl-a ($A$) and two parameters, the growth rate $r$ and the carrying capacity $K$:

$$\frac{dA}{dt} = rA \times \left(1 - \frac{A}{K}\right)$$

The analytical solution: $A_t = \frac{A_0 \cdot K}{A_0 + (K - A_0) \cdot \exp(-r \cdot t)}$ was added as user-defined equation to the Dynamic Fit Wizard in the program Sigma Plot, version 14.0, and growth rates were determined by iterative non-linear regression. Growth rates were compared by one-way ANOVA for TRW and WC medium, as no reliable estimates could be obtained for RRW, and therefore the preferred test, a two-way ANOVA with cultures inoculated and medium type (TRW, RRW, WC) as fixed factors, was not performed.

## 3. Results

### 3.1. Determination of optimal dose of flocculants

The turbidity of the reservoir combined with different coagulants (chitosan, PAC, MOS and a combination of MOS and chitosan) resulted in the settling of cyanobacteria after 1h treatment period (Fig 2).

In the chitosan series, a concentration of 1 and 2 mg/L reduced Chl-a in the top of the tubes by 13% and 23%, respectively. Chitosan doses of 4 and 8 mg/L lowered the Chl-a concentration in the top of the tubes by 59.5% and 86.1%, respectively, while in the bottom of the tubes, the Chl-a concentration increased 6.2 and 8.2 times respectively (Fig 2A).

At all doses of chitosan, pH and PSII dropped only gradually. In the top of the tubes, turbidity was reduced by 40.5% and 43.5% at doses of 1 mg/L and 2 mg/L chitosan, respectively.

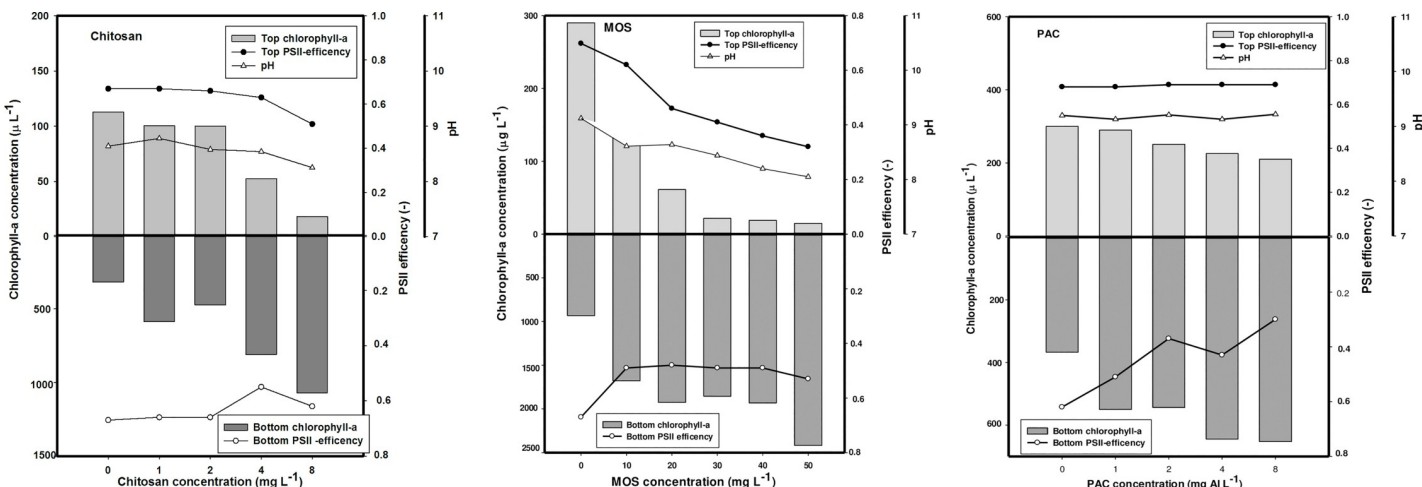

**Fig 2. Cyanobacterial removal efficiency of different flocculants in floc and sink experiment.** Chlorophyll-*a* concentrations (μg/ L) in the top 2 mL (top light gray bars) and bottom 2 mL (lower dark gray bars) of cyanobacteria suspensions incubated for 1 h with different concentrations of flocculants: A) chitosan B) MOS & C) PAC. For chitosan, MOS and PAC series, open triangle represents the pH value. Closed and open circle represents the top and bottom PSII efficiency values.

While, at a doses of 4 mg/L and 8 mg/L chitosan, the turbidity was reduced by 62.1% and 77.1%, respectively. In contrast, at the bottom of the tubes, at chitosan doses of 4 mg/L and 8 mg/L the turbidity was increased 2.4 and 3.5 times, respectively (Fig 3A). Hence, chitosan concentration of 4 mg/L was considered the effective dose without affecting the pH and PSII.

In the MOS series, doses of 20 and 30 mg/L reduced Chl-a sharply in the top of the tube by 79% and 93%, respectively. At the top of the tube, MOS concentrations of 40 mg/L and 50 mg/L reduced Chl-a further to 92.6% and 95%, respectively. In contrast, at the bottom of the tubes, the MOS doses of 30 mg/L, 40 mg/L, and 50 mg/L increased Chl-a levels 2.7, 2.9, and 3.5 times (Fig 2B). At 20 mg/L MOS, PSII was reduced from 0.52 to 0.35, at 30 mg/L then afterward, remained around 0.35, while PSII dropped further to 0.26 and 0.23 at doses of 40 mg/L and 50 mg/L respectively. The pH dropped only gradually from pH 9.12 to pH 8.05 (Fig 2B).

In the top of the tubes, MOS concentration of 20 mg/L and 30 mg/L reduced the turbidity by 67% and 79%, respectively, whereas in the bottom of the tubes, turbidity increased by 6 and 6.8 times, respectively (Fig 3B). Therefore, MOS concentration of 30 mg/L was considered the effective dose without affecting the pH and PSII.

In the PAC series, at concentrations of 1 and 2 mg Al/L, Chl-a in the top of the tubes remained unaffected. PAC concentrations of 4 and 8 mg Al/L, reduced Chl-a concentrations in the top of the tubes by 24% and 30%, respectively, while in the bottom of the tubes, the Chl-a levels increased 1.2 and 1.4 times (Fig 2C). At PAC dose of 1 mg Al/L, PSII efficiency dropped from the control (0.79) to 0.64, whereas at PAC dose of 2, 4 and 8 mg Al/L, PSII efficiency remained unaffected, indicating that the performance of cyanobacteria was not affected by increasing PAC doses with pH remaining unaffected at all doses.

In the top of the tubes, PAC doses of 4 mg Al/L and 8 mg Al/L reduced the turbidity by 52% and 56%, respectively, while in the bottom of the tubes, the turbidity increased by 1.7 and 2.0 times, respectively (Fig 3C). Hence, PAC concentration of 4 mg Al/L was considered an effective dose without affecting the pH and PSII efficiency.

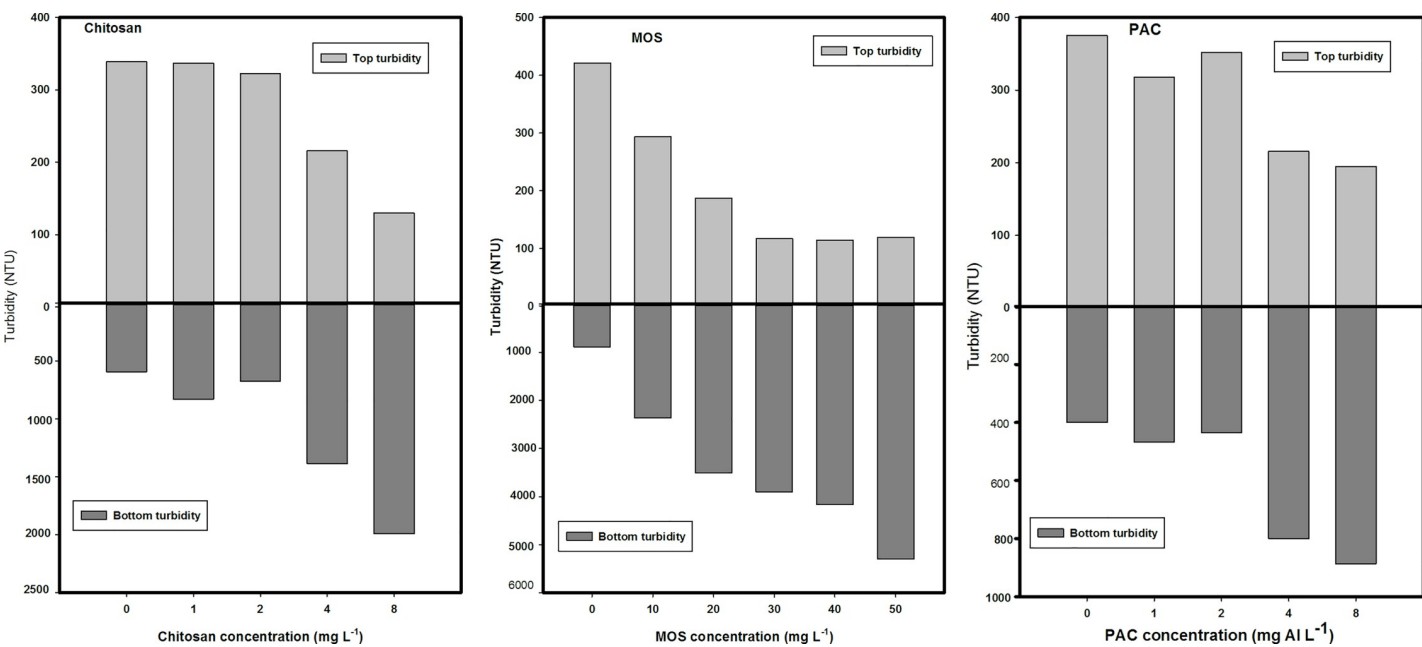

**Fig 3. Effect of different flocculants on reservoir turbidity.** Level of turbidity (NTU) in the top 2 mL and bottom 2 mL of cyanobacterial suspensions incubated for 1 h with different concentrations of flocculants: A) chitosan top light gray chitosan bottom dark gray B. MOS top light gray and MOS bottom dark gray C) PAC top light gray and PAC bottom dark gray.

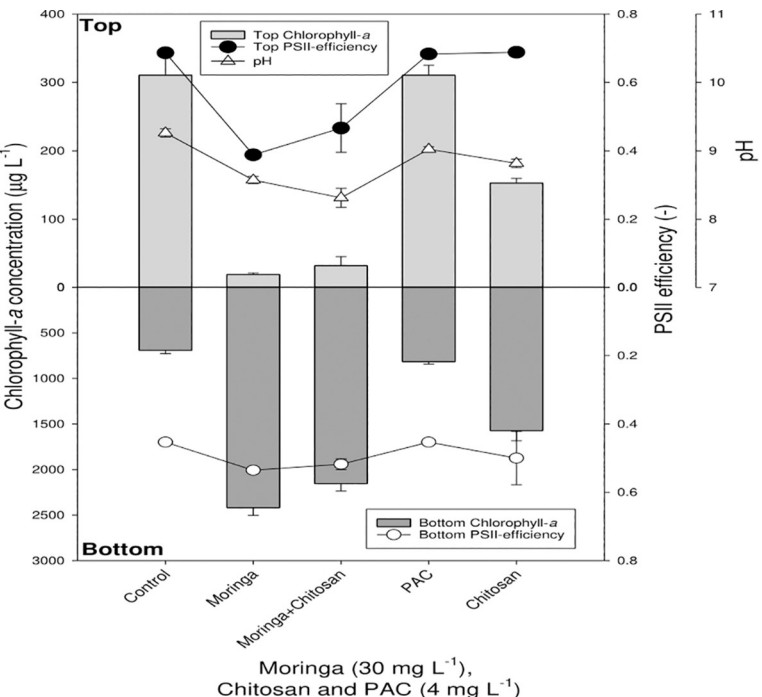

**Fig 4. The clearing effect of flocculants on the reservoir water.** Chlorophyll-*a* concentrations (μg/ L) in the top 2 mL (top light gray bars) and bottom 2 mL (lower dark gray bars) of 130 μg/L cyanobacterial suspensions incubated for 1 h with different concentrations of the flocculants: PAC (4 mg Al/ L), chitosan (4 mg/L) and MOS (30 mg/L). Also included are the pH values (open triangles), PSII efficiency in the top (closed circles) and bottom (open circles) of the experimental tubes.

## 3.2. The efficacy of coagulants in removing cyanobacteria from the reservoir water

At 30 mg/L MOS, most of the cyanobacterial cells were forced to sink to the bottom of the tubes, while the top layer of the tubes became clear (Fig 4). MOS was the most effective coagulant. One-way ANOVA indicated that Chl-a concentrations among treatments were significantly different for both the top ($F_{4,14}$ = 179.8; $p < 0.001$) and the bottom ($F_{4,14}$ = 323.0; $p < 0.001$) of the test tubes. Likewise, the combination of 4 mg/L chitosan and 30 mg/L MOS was highly efficient in clearing the top water layer and settling most of the cyanobacteria to the bottom of the tubes. Compared to other coagulants, PAC showed the least cyanobacteria removal efficiency (Fig 4).

PSII efficiency remained fairly stable in all treatments (chitosan, MOS and PAC) in both the top and bottom of the test tubes. Kruskal-Wallis One Way Analysis of Variance on Ranks indicated that differences in PSII among treatments were significant for the top ($H_4$ = 11.249; $p = 0.024$), but not for the bottom ($H_4$ = 8.841; $p = 0.065$) of the test tubes. The pH showed only minor fluctuations between pH 6.7 and 7.3 (Fig 4).

## 3.3 The effect of treated reservoir water on cyanobacterial re-growth

The median Chl-a concentrations of different cyanobacterial species (*Anabaena flos-aquae*, *Microcystis aeruginosa*, and *Microcystis* sp. culture from Legedadi Reservoir) grown in different media are shown in Table 1.

Overall, cyanobacterial species did not show any sign of growth during the first 3 days of the experimental period. In TRW, all cultures showed a period of about 10 days with slow

**Table 1. Variations in chlorophyll-a (µg L$^{-1}$) levels among experimental treatments.**

| Treatment | Median | Interquartile range (IQR) |
|---|---|---|
| TRW culture(*Microcystis* sp.) | 61.4 | 19.5–1888.9 |
| TRW *Anabaena flos-aquae* | 57.8 | 20–1336.2 |
| TRW *Microcystis aeruginosa* | 35.6 | 18–1218.3 |
| Modified WC culture(*Microcystis* sp.) | 306.5 | 37.6–554.4 |
| Modified WC *Anabaena flos-aquae* | 46.6 | 11.9–696.7 |
| Modified WC *Microcystis aeruginosa* | 111.5 | 25.6–1446.7 |
| RRW culture(*Microcystis* sp.) | 389.3 | 175.9–494.4 |
| RRW *Anabaena flos-aquae* | 382.3 | 170–557.1 |
| RRW *Microcystis aeruginosa* | 342.6 | 162.3–726.5 |

growth, where after growth accelerated (Fig 5A). One-way ANOVA indicated no differences in growth rates among cultures ($F_{2,8}$ = 1.653; $p$ = 0.268). The overall average growth rate was 0.39 (± 0.16)/day. In WC medium, the cyanobacteria started growing earlier than in TRW, but did not reach the high biomass as in TRW after 26 days (Fig 5B). The cultures with *Microcystis* PCC7820 added (labelled '*Microcystis*') reached carrying capacity after 14 days, *Anabaena* started to decline after 14 days, while *Microcystis* from Legedadi (labelled 'Culture') started with slower growth, but reached higher Chl-a concentration at the end of the experiment (Fig 5B). One-way ANOVA indicated significant differences in growth rates among cultures ($F_{2,7}$ = 29.839; $p$ = 0.002). All Pairwise Multiple Comparison Procedures (Holm-Sidak method) revealed that growth rates of all cultures were significantly different from each other ($p < 0.05$). Growth rate of the culture was highest with 1.30 (± 0.24)/day, that of *Microcystis* PCC7820 was 0.64 (± 0.16)/day and of the *Anabaena* cultures it was 0.25 (± 0.6)/day. Iterative regression on data of one of the Legedadi *Microcystis* incubations did not converge ($r^2$ = 0.186), hence, no growth rate of that replicate was determined. In RRW, all incubations showed a decline in Chl-a after 14 days. The logistic model only yielded poor fits ($r^2$ = 0.131–0.640) and no reliable growth rates could be determined (Fig 5C). The RRW without any cyanobacteria added showed a growth rate of 0.22/day ($r^2$ = 0.940).

## 4. Discussion

### 4.1. The effect of coagulants/flocculants on the cyanobacteria

The results of this study are in line with the hypothesis that adding coagulants to water from the turbid Legedadi Reservoir could effectively remove cyanobacteria and suspended solids from the water without the need for adding additional ballast material. However, this water clearing effect was only observed when chitosan, MOS or their combination was used, but rather unexpectedly, not with PAC.

In this study, PAC doses of 4 and 8 mg Al/L were ineffective in coagulating and settling the cyanobacterial species in the reservoir's water samples. In contrast, some previous studies showed that even lower doses of PAC (1 and 4 mg Al/L) effectively formed flocs of buoyant cyanobacteria [20,28]. The most likely reason for the low coagulation efficiency of PAC in the current study could be the relatively high pH (9.12–9.24) of the water. The observed higher pH was a physical state commonly associated with cyanobacterial blooms [29]. A high pH, along with high alkalinity may hinder the coagulating efficiency of PAC, thereby affecting its ability to effectively remove cyanobacterial species from the water bodies [30]. A similar observation was made by [19], who observed that large flocs were only formed at PAC doses of > 8 mg Al/L in water that had a high pH (pH = 10), as those higher PAC doses were able to reduce the pH

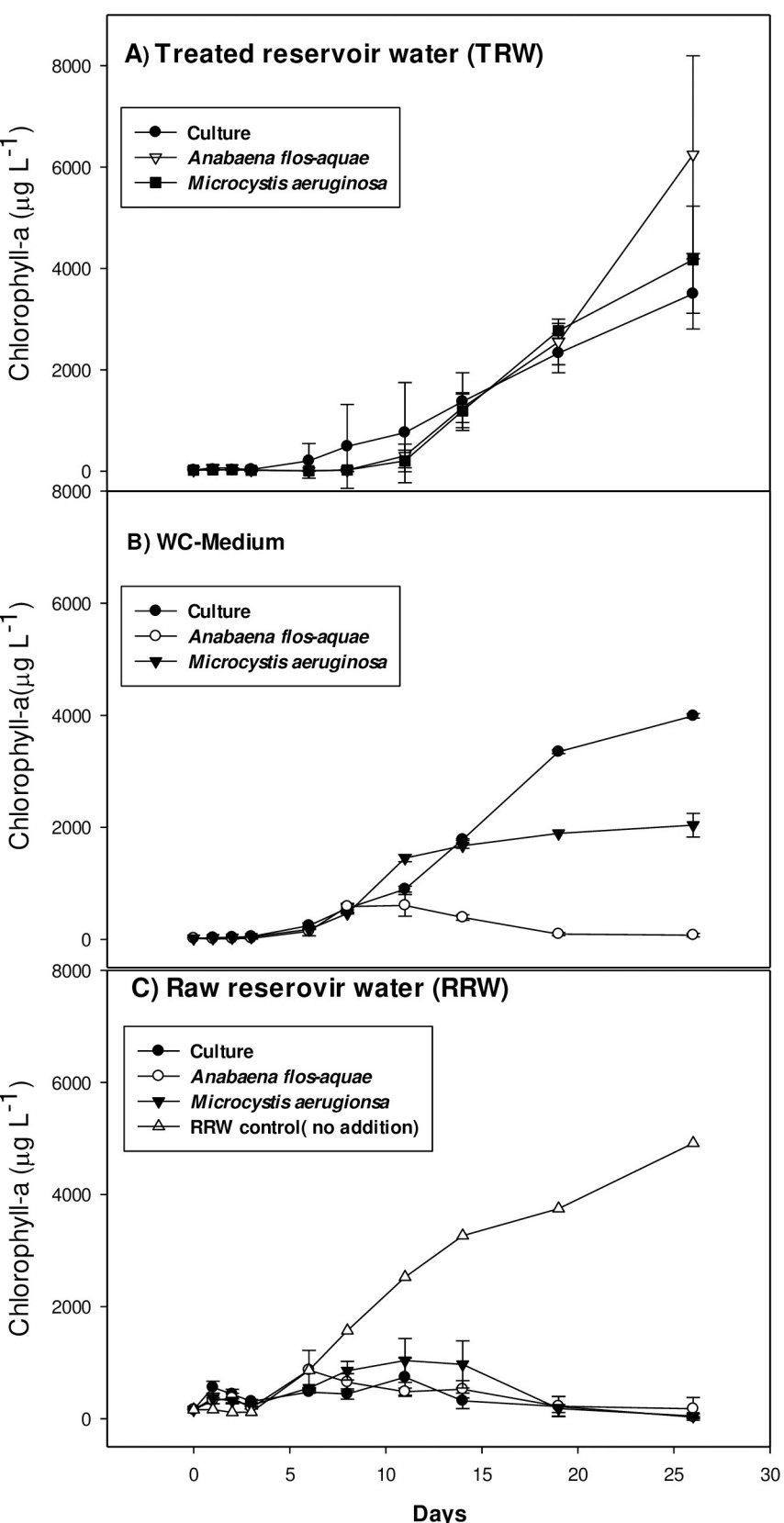

**Fig 5. Growth of cyanobacterial species in different medium.** *Microcystis* sp.culture from Legedadi Reservoir (Culture), *Anabaena flos-aquae*, and *Microcystis aeruginosa* in different media incubated for one month in A, Treated Reservoir Water (TRW) B, modified WC growth medium, and C, Raw Reservoir Water (RRW).

of the water to levels that allowed formation of aluminium hydroxide flocs. Hence, it is likely that higher PAC doses of > 8 mg Al/L could also have been effective in Legedadi Reservoir water. However, due to potential health concerns of aluminium in finished drinking water, the World Health Organization (WHO) recommends to minimize aluminium levels in water to 0.1–0.2 mg/L [31]. Thus, additional experiments can be recommended to test PAC coagulation at a higher dose, but those should also include estimates of residual, dissolved organically bound Al [32].

Organic coagulants, such as chitosan, are viewed as safer, non-toxic and eco-friendlier coagulants than metal-based coagulants like PAC [33]. Different studies have revealed that chitosan is an effective bio-coagulant used in the removal of cyanobacteria from water [19,20,24]. In Legedadi Reservoir water, chitosan removed cyanobacterial species effectively at a dose of 8 mg/L despite the prevalence of moderately high pH. Some studies have reported that pH influences chitosan coagulation properties [34,35], and that high pH hampers the coagulation [36]. In this study, pH was not above 9 in the chitosan treatments, and at a dose of 8 mg/L pH was even reduced to almost 8. Clearly, pH was somewhat lower than that in the PAC treatment (see Fig 2), which urges for care in comparing the efficacy of the different coagulants. Chitosan may also cause cell lysis in cyanobacteria [37,38], therewith releasing cyanotoxins to the water [19]. In our study, no indications for rapid cell lysis were observed, no increase in chlorophyll-a signal was detected, and no strong decline in PSII efficiency to value of or close to zero were measured [38].

The other natural coagulant used in this study, *Moringa oleifera* seed extract (MOS) also removed turbidity and cyanobacteria from the reservoir water. This is in agreement with other studies that showed MOS was effective in flocculating and removing cyanobacteria from water [24,39]. MOS, dosed at 50 mg/L, could reduce up to 90% of the turbidity and the *Microcystis aeruginosa* Chl-a concentration [40]. MOS along with chitosan removed *M. aeruginosa* cells effectively in freshwater, and *Amphidinium carterae* and *Chlorella* sp. in seawater [24]. Other studies also reported MOS efficiency in removing harmful species of *Microcystis* [25,41]. In the current experiment, 30 mg/L MOS was effective in removing cyanobacterial species without observable cell lysis as PSII efficiency was only marginally affected.

This effective dose of MOS is relatively lower than the doses used in several other studies e.g. [40,42] due to the high turbidity of the reservoir. According to [40,42] the optimum dose of MOS is reduced as the water turbidity increases. The ability of MOS to flocculate the algal cells is ascribed to the interaction between the cells and the active protein molecule in the MOS. The cyanobacteria behave like negatively charged particles because of chemical materials comprising the cell walls, while the active cationic protein molecule in the MOS acts as positively charged particle that interacts with the cell wall and flocculated the cyanobacteria [43].

At higher concentrations of 50 mg/L, MOS and above, there was a sign of potential cell lysis with a stronger reduction of PSII efficiency. When a considerable proportion of the cells are lysed, a sharp drop in PSII-efficiency could be expected [44]. Therefore, using MOS at lower dose is crucial since MOS is known to be (more) effective in removing cyanobacteria, but less effective in removing dissolved toxins such as microcystins [42], although a recent study found up to almost 55% removal of microcystins by MOS [41].

The observed decline in PSII efficiencies indicated that there is some detrimental effect of MOS on the cyanobacteria occurring in the relatively short duration of the experiments.

However, previous growth experiments with MOS have revealed that growth of *M. aeruginosa* is not hampered at 32 mg/L, while it was strongly reduced at 64 mg/L [45]. Together with findings of [46], who reported that at doses of 20–160 mg/L MOS, *M. aeruginosa* populations died, which was corroborated by a rapid drop in PSII efficiency to zero. Higher dose of MOS is not recommended. At high dose, the flocculation effect of MOS extract is reduced owing to charge neutralization, which results in charge reversal and destabilization of the cyanobacterial cells [39]. Some potential damage, causing a delayed effect on *M. aeruginosa* in settled flocs, has been observed for chitosan [47]. Such delayed effect can be viewed as beneficial, because damaging and gradual lysis of settled cyanobacteria (within a few days) will minimize the risk of resuspension and recolonization of the water column. Moreover, cyanotoxins will be degraded rapidly by the decomposing bacteria living near the sediment [48].

The relatively high pH observed in this experiment did not show major influence on the flocculation capacity of MOS, which is supported by findings in other studies that also concluded cyanobacterial removal efficiency of MOS is not affected by pH in the range of pH 5–9 [25,49]. Hence, MOS could be a good alternative for coagulating and settling cyanobacteria in water bodies that exhibit high pH. However, as the isoelectric point of the coagulation proteins of MOS is around pH 10 [50], pH 10 seems to be the upper limit of MOS use.

## 4.2. Floc and sink techniques of cyanobacteria removal

The results of this study show that in tropical turbid and moderately high pH reservoirs/lakes, low- dose biodegradable and environmentally safe coagulants such as, MOS, chitosan or a combination MOS, and chitosan is a good alternative for effectively settling harmful cyanobacteria out of the water column. The results also indicated that the turbidity of the reservoir, which is a result mostly of suspended solids, could be a good substitute for external ballast material in settling cyanobacteria.

Previous studies used a 'floc and sink' technique in which ballast material was introduced before the addition of an effective dose of coagulant [13,14,19,20,24,26,51].

The rational of testing coagulants using the reservoir turbidity as ballast in floc and sink techniques has several advantages: 1) restoration cost would be cheaper as it will reduce the cost of adding external ballast material, 2) the turbidity will be reduced, and the reservoir water will regain its transparency allowing increased resilience against disturbances, and 3) drinking water treatment cost will be reduced.

The fact that this experiment was conducted using reservoir water samples makes the results of this investigation more realistic. The present results are important in combating positively buoyant, surface scum-forming cyanobacteria that present one of the greatest threats to the well-being of humans and animals [52]. Besides, intervention strategies like using algaecides that cause liberation of toxins from cells are not the preferred management options as they eventually lead to the exposure of the public to cyanotoxin [9]. Thus, water supply managers should reconsider the use of algaecides in drinking water source water bodies and replace them with effective and environmentally safe natural flocculants like MOS. MOS can also reduce the concentration of extracellular cyanotoxins (microcystins) by 50% [41]. However, there are concerns regarding the use of biodegradable flocculants like MOS. MOS availability is limited to certain regions with tropical and subtropical climates [53]. Furthermore, *Moringa oleifera* is a multipurpose tree. Thus, its use in the removal of cyanobacteria leads to a conflict of interest. Besides, applying MOS in the reservoir might increase the organic load of the reservoir, thereby affecting the level of DO [54]. Moreover, high doses of MOS not only increase organic carbon, but also nutrients that can promote microorganisms growth and cause odour, taste, and colour problems [55,56].

Therefore, applying a minimum dose of MOS is preferable. MOS combined with chitosan will reduce the amount of MOS needed for water restoration. Therefore, MOS combined with chitosan can be a good alternative coagulant for reservoirs/lakes in the tropical region and more up-scaled experiments can be considered.

### 4.3 Effect of *Moringa oleifera* in removing and preventing re-growth of cyanobacteria

The results of our growth experiments indicated that the low dose of MOS is not only efficient in removing buoyant cyanobacteria from the reservoir water, it also delayed the re-growth of common bloom-forming cyanobacterial species such as *Microcystis aeruginosa* and *Anabaena flos-aquae*. During the first 3 days of the experimental period, the growth of the species in all media was negligible. This delay in cyanobacterial growth could be due to the need for acclimatization through biochemical adjustment [57]. The prolonged acclimatization in TRW (indicated with "TRW" in Fig 5) was unexpected and in contrast to our hypothesis as in the new clear water state, nutrients were believed to enhance the growth of cyanobacteria (20 μg/L) inoculum. The delay could be a result in of inhibitory factors or anti-cyanobacterial action of MOS [46]. However, within the course of the experiment, cyanobacterial Chl-a concentrations reached much higher values in TRW than in either the raw reservoir water (RRW) or in the artificial growth medium (WC medium). A higher biomass of carrying capacity in the TRW treatment may be caused by additional nutrients added from MOS, which contains phosphate, nitrate, ammonium [46]. The much lower growth and even decline after 14 days observed in RRW could have been caused by higher turbidity, yet the single RW control seems to contradict this. The experiment cannot provide an answer why Chl-a in the RW was much higher than in RW to which cyanobacteria were added (RRW treatment). Clearly, more research will be needed, nonetheless, the experiment provides indications that clearing the reservoir with ongoing nutrient inputs may come with a risk that cyanobacterial biomass will reach much higher concentrations after in-situ coagulation and sinking than without. Up-scaled experiments are needed to get insight in the longevity of the intervention and possible drawbacks as outlined above.

## 5. Conclusions

- These experimental results indicate the potential applicability of *Moringa oleifera* seed or a combination of *Moringa oleifera* seed and chitosan as flocculants without adding ballast material in removing cyanobacterial species in turbid lakes/reservoirs.

- *Moringa oleifera* seed seems to cause a short-term delay in re-growth of cyanobacteria after treatment of tropical lakes/reservoirs.

- Floc and sink experiments were conducted effectively in a small sample volume (11 mL) of reservoir/lake water.

- Up-scaled experiments are needed.

## Supporting information

**S1 Fig.** Experimental set-up of the Floc and Sink technique: (A) conceptual diagram, (B) Coagulation experiment with different coagulants (C) Top and bottom 2mL samples after 1hr experiment.
(TIF)

**S1 Table. One way ANOVA analysis on effect of coagulants on cyanobacterial species.**
(DOCX)

**S1 File. Descriptive statistic of growth of cyanobacterial species on different growth medium.**
(DOCX)

**S2 File. Iterative non-linear regression analysis of cyanobacterial growth at different medium using a logistic growth model.**
(XLSX)

## Acknowledgments

We thank Addis Ababa water and sanitation authority for assistance during sampling time. We also thank the Aquatic Ecology and Water Quality Management group at Wageningen University for providing all the laboratory facilities.

## Author Contributions

**Conceptualization:** Hanna Habtemariam, Miquel Lürling.

**Data curation:** Hanna Habtemariam.

**Formal analysis:** Hanna Habtemariam, Demeke Kifle, Miquel Lürling.

**Investigation:** Hanna Habtemariam, Maíra Mucci.

**Resources:** Miquel Lürling.

**Supervision:** Demeke Kifle, Seyoum Leta, Miquel Lürling.

**Writing – original draft:** Hanna Habtemariam, Demeke Kifle.

**Writing – review & editing:** Seyoum Leta, Miquel Lürling.

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
