## [Decision Letter · Decision Letter 0]

15 Dec 2020

PONE-D-20-31065

Removal of cyanobacteria from a water supply reservoir by sedimentation using natural flocculants and water turbidity as ballast: Case of Legedadi Reservoir (Ethiopia)

PLOS ONE

Dear Dr. Habtemariam,

Thank you for submitting your manuscript to PLOS ONE. After careful consideration, we feel that it has merit but does not fully meet PLOS ONE’s publication criteria as it currently stands. Therefore, we invite you to submit a revised version of the manuscript that addresses the points raised during the review process. In particular, the first reviewer has made several criticisms on some methodological choices that should be addressed in your revision version. In addition to that, I want also to invite you to be more cautionous in the discussion, in the extrapolation of your experimental findings to the real conditions in the lake.

We look forward to receiving your revised manuscript.

Kind regards,

Jean-François Humbert

Academic Editor

PLOS ONE

Journal Requirements:

'Funding

This research did not receive any specific grant from funding agencies in the public, commercial, or not for profit sectors.'

'Hanna habtemariam

Demeke Kifle

Seyoum Leta

Maíra Mucci

Miquel Lürling '

'Hanna habtemariam

Demeke Kifle

Seyoum Leta

Maíra Mucci

Miquel Lürling '

a. Please complete your Competing Interests statement to state any Competing Interests.

If you have no competing interests, please state "The authors have declared that no competing interests exist.", as detailed online in our guide for authors at http://journals.plos.org/plosone/s/submit-now

4. We note that Figure 1 in your submission contains map images which may be copyrighted.

We require you to either (a) present written permission from the copyright holder to publish these figure specifically under the CC BY 4.0 license, or (b) remove the figure from your submission:

b. If you are unable to obtain permission from the original copyright holder to publish these figure under the CC BY 4.0 license or if the copyright holder’s requirements are incompatible with the CC BY 4.0 license, please either i) remove the figure or ii) supply a replacement figure that complies with the CC BY 4.0 license. Please check copyright information on all replacement figures and update the figure caption with source information. If applicable, please specify in the figure caption text when a figure is similar but not identical to the original image and is therefore for illustrative purposes only.

5. Please upload a new copy of Figure 2a as the detail is not clear. Please follow the link for more information: https://blogs.plos.org/plos/2019/06/looking-good-tips-for-creating-your-plos-figures-graphics/

6. Please include a copy of Table 1 which you refer to in your text on page 13.

Reviewers' comments:

Reviewer's Responses to Questions

**Comments to the Author**

1. Is the manuscript technically sound, and do the data support the conclusions?

Reviewer #1: Partly

Reviewer #2: Yes

2. Has the statistical analysis been performed appropriately and rigorously? 

Reviewer #1: No

Reviewer #2: Yes

3. Have the authors made all data underlying the findings in their manuscript fully available?

Reviewer #1: Yes

Reviewer #2: Yes

4. Is the manuscript presented in an intelligible fashion and written in standard English?

Reviewer #1: Yes

Reviewer #2: No

5. Review Comments to the Author

Reviewer #1: This study investigated the removal of cyanobacteria from a water supply resevoir by sedimentation using natural flocculants and water turbidity as ballast. This is a meaningful study, however, this manuscript is still not in a publishable state. The manuscript needs to be further improved in research design and analysis. And some sentences in this manuscript are hard to understand, it needs careful editing by someone with expertise in technical English editing paying particular attention to English grammar, spelling, and sentence structure so that the goals and results of the study are clear. Furthermore, some important aspects of this study are confusing.

1. Title: “use water turbidity as ballast”, but there’s no experimental design on how to use the turbidity of water as the ballast in this paper.

2. Grammar check: such as page 5, line 88-90; page 7, line 139-141……

3. Page 5, line 90-91: “Legedadi Reservoir is a very turbid reservoir”, how high is the turbidity? “the high turbidity was considered as a unique feature and expected to act as ballast”, does the high turbidity reservoir only be used as a case study? It should be designed to compare the effects of different turbidity.

4. Section 2.1: The water quality of the reservoir is not explained. Water quality is an important condition for the research, such as the turbidity of the reservoir when the experiment was carried out, the dominant algae and algae concentration, etc.

5. Page 7, line 149: “After dosing, the contents in each test tube were mixed briefly using a glass rod.” It is confusing. Coagulation-flocculation experiments were usually performed by jar test with a six-paddle stirrer. How to control the specific conditions of the mixing operation using a glass rod, so whether the experimental results of the glass rod mixing would be affected by the man-made operation errors?

6. Page 8, line 161-163: The dose design of PAC, Chitosan, and MOS was different, would the difference of coagulation effect be affected by the dosage range of different coagulants? For example, the dose of 8mg/L chitosan as shown in Fig. 3 has not been proved to reach the optimal dose with stable removal efficiency, while the span of MOS dose is larger with 20 mg/L has a relatively high efficiency and 30mg/L has reached the optimal dose.

7. Section 2.4: It is not recommended to use question sentence as subtitle in the experimental methods section.

8. Page 11, line 233-234: What does the “drop” and “remian” of PSII indicate? It should be explained.

9. Page 12, line 253: It is recommended to list the process and results of data analysis in a table or supporting information, and the supporting information in this paper seems to have uploaded the wrong file.

10. Page 13, line 288: The “Linear regression model” is recommended to provide in detail.

11. Page 14, line 297: “8Al/L”, check unit.

12. Page 15, line 330-331: “At higher concentrations of 50 mg/L MOS and above, there was a sign of cell lyses with a high reduction of PSII efficiency”. However, Fig. 3 showed that PSII efficiency had dropped when the MOS dose is 20 mg/L. Confusing.

13. Page 16, line 345: Section 4.2, check subtitle.

14. Fig. 2 is not clear, and there is no directional significance of the actual results. What is the author’s intention to put Fig. 2 ?

15. Fig. 3, “Moringa” in Fig.3B should be consistent with “MOS” in the text.

16. Fig.4, why not put MOS results in the same figure with PAC and chitosan? Using the drawing method of Fig. 3 to distinguish the top and bottom is a better choice.

17. Fig.6 is not clear.

18. References: It would be better to quote more recent references such as in the last three years.

Reviewer #2: The manuscript evaluated the efficiency of the flocculants/coagulants chitosan, Moringa oleifera seed (MOS) and poly-aluminium chloride (PAC) in settling cyanobacterial species in Legedadi Reservoir. The authors suggest that the efficacy of a flocculant/coagulant in the removal of cyanobacteria is influenced by the uniqueness individual lakes/reservoirs, implying that mitigation methods need to consider the unique characteristic of the lake/reservoir.

The manuscript was generally well organized, and the conclusion is constructive to remove cyanobacteria and control the blooms.

Comments for revision and correction:

1) Introduction: the authors give more information and description to support their assumption: water turbidity functions as ballast.

2) The chemical composition of Moringa oleifera seed may change between those collected from different locations. This information is helpful to understand its efficiency.

3) In Fig.3 A &C，Chla concentration was higher in the high treatments than in the control for bottom and surface?

4) In Fig.5，treated with MOS，PSII efficiency decreased at the surface, but increased for cyanobacteria at the bottom. Why?

5) As turbidity is assumed to be ballast, it is better to give more physical and chemical information of the turbidity. Know which kinds of turbidity function better.

6) In Fig.3, Fig.4, Fig.7, adding error bars.

7) Redrawing Fig.6 to be clear for readers.

8) The English needs correction. For example, Line 86: future =feature? Line 88-89: “The availability of local soils with natural P binding capacities was. For

instance, considered…” =” For instance, the availability of local soils with natural P binding capacities was considered… ”.

6. PLOS authors have the option to publish the peer review history of their article (what does this mean?). If published, this will include your full peer review and any attached files.

Reviewer #1: No

Reviewer #2: No

---

## [Author Response · Author response to Decision Letter 0]

8 Feb 2021

Responses to editorial requirements

COMMENT

Please ensure that your manuscript meets PLOS ONE's style requirements, including those for file naming

RESPONSE 

All contents of the manuscript is revised according to PLOS ONE's style

COMMENT

We note that you have provided funding information that is not currently declared in your Funding Statement. However, funding information should not appear in the Acknowledgments section or other areas of your manuscript. Please remove any funding-related text from the manuscript and let us know how you would like to update your Funding Statement. Currently, your Funding Statement reads as follows:

'Hanna habtemariam

Demeke Kifle

Seyoum Leta

Maíra Mucci

Miquel Lürling '

RESPONSE

Thank you for your comment. We removed funding related text information from the manuscript. Additional we specify our funding statements on the cover page

COMMENT

Please complete your Competing Interests statement to state any Competing Interests a. Please complete your Competing Interests statement to state any Competing Interests. If you have no competing interests, please state "The authors have declared that no competing interests exist.", as detailed online in our guide for authors at http://journals.plos.org/plosone/s/submit-now

RESPONSE

Thank you for your comment and guidance we have included our Competing Interests statement on the cover letter

COMMENT

4. We note that Figure 1 in your submission contains map images which may be copyrighted. We require you to either (a) present written permission from the copyright holder to publish these figure specifically under the CC BY 4.0 license, or (b) remove the figure from your submission:

If you are unable to obtain permission please either i) remove the figure or ii) supply a replacement figure that complies with the CC BY 4.0 license. Please check copyright information on all replacement figures and update the figure caption with source information. If applicable, please specify in the figure caption text when a figure is similar but not identical to the original image and is therefore for illustrative purposes only.

RESPONSE

Thank you for your comment. we removed figure 1 from the manuscript and replaced with new Figure

COMMENT

5. Please upload a new copy of Figure 2a as the detail is not clear. Please follow the link for more information: https://blogs.plos.org/plos/2019/06/looking-good-tips-for-creating-your-plos-figures-graphics/

RESPONSE

Thank you for the comment we have uploaded a copy of figure 2a (in tiff format)

COMMENT

6. Please include a copy of Table 1 which you refer to in your text on page 13.

RESPONSE

Thank you for your comment. We have included Table 1 on page 13 line 299

Responses to Reviewer’s comments

Reviewer#1

COMMENT

Further improved in research design and analysis. And some sentences in this manuscript are hard to understand, it needs careful editing by someone with expertise in technical English editing paying particular attention to English grammar, spelling, and sentence structure so that the goals and results of the study are clear some important aspects of this study are confusing.

RESPONSE 

Thank you for your critical review. We revised the entire manuscript with correction of grammar, spelling and by improving readability. The changes made in the manuscript were red-highlighted

COMMENT

Title: “use water turbidity as ballast”, but there’s no experimental design on how to use the turbidity of water as the ballast in this paper.

RESPONSE

Thank you for your comment. In the ‘floc and sink’ approach a ballast compound is first introduced where after flocculants/coagulants will be added to settle phytoplankton out of the water. In the current experiment, since the reservoir is turbid, because of high concentrations of suspended solids, we hypothesized that there would be no need of adding external ballast. Therefore, we designed an experiment applying only coagulants/flocculants, which were added to the reservoir water without any additional ballast. We agree that the term “water turbidity “ is not clear and hence we have rephrased the title emphasizing “suspended solids. We also have removed “natural” from the title as PAC cannot be viewed as a natural coagulant. The new title reads: “Removal of cyanobacteria from a water supply reservoir by sedimentation using flocculants and suspended solids as ballast: Case of Legedadi Reservoir (Ethiopia)” 

COMMENT

Grammar check: such as page 5, line 88-90; page 7, line 139-140

RESPONSE

We accepted the comment and corrected the grammar on page 5, lines 85-91; page 7, lines 155-157

COMMENT

Page 5, line 90-91: “Legedadi Reservoir is a very turbid reservoir”, how high is the turbidity? “the high turbidity was considered as a unique feature and expected to act as ballast”, does the high turbidity reservoir only be used as a case study? It should be designed to compare the effects of different turbidity.

RESPONSE

We appreciated the comments provided by the reviewer. The annual mean turbidity of the reservoir was mentioned on page 7 line 142 of the original version of the manuscript. The turbidity in the reservoir is high with an annual mean turbidity of 433 NTU, and during the rainy season increasing up to 800–900 NTU. This is mostly due to suspended solids flowing in from the rivers and considered high enough to form easy coagulation with coagulants like MOS. Inasmuch as we’ve performed the experiments at Wageningen University, we were limited in the amount of Reservoir Water to be used. Indeed, with appropriate equipment it would be very nice to have a series of those experiments performed over the course of the year, since several water quality variables will change that could have an effect on removal efficiency.

COMMENT

Section 2.1: The water quality of the reservoir is not explained. Water quality is an important condition for the research, such as the turbidity of the reservoir when the experiment was carried out, the dominant algae and algae concentration, etc.

RESPONSE

We highly appreciate the comments. We have now included the concentrations of nutrients, pH, conductivity, and dominant algal groups encountered during the sampling time in section 2.1 (page 6, line 120–127). Information regarding algal concentration was included in the original version of the manuscript on page 8, line 158, while level of turbidity was indicated on page 7 line 153.

COMMENT

Page 7, line 149: “After dosing, the contents in each test tube were mixed briefly using a glass rod.” It is confusing. Coagulation-flocculation experiments were usually performed by jar test with a six-paddle stirrer. How to control the specific conditions of the mixing operation using a glass rod, so whether the experimental results of the glass rod mixing would be affected by the man-made operation errors?

RESPONSE

Thank you very much for your concern and comment. It is true that mixing is an important component of the coagulation process and in several coagulation studies, it was done in a jar with stirrer. However, in our experiment, we chose to opt for a more gentle mixing as has been done in most ‘floc and sink’ studies (e.g. Noyma et al., 2016, 2017; Miranda et al., 2017; de Lucena-Silva et al., 2019; Lürling et al., 2020). The rationale behind glass-rod stirring is that this mixing is more representative to the gentle mixing that often occurs in situ, while the standard jar test includes periods of vigorous mixing. Previous studies have shown that “mixing regime is not that crucial for flocs formation as long as there is slow mixing. Such mixing will also be the predominant regime in whole lake treatments” (Lürling et al., 2017).

de Lucena-Silva, D., Molozzi, J., dos Santos Severiano, J., Becker, V., de Lucena Barbosa, J.E., 2019. Removal efficiency of phosphorus, cyanobacteria and cyanotoxins by the “flock & sink” mitigation technique in semi-arid eutrophic waters. Wat. Res. 159: 262-273.

Lürling, M., Noyma, N., deMagalhães, L., Miranda, M., Mucci, M., van Oosterhout, F., Huszar, V.L., Marinho, M.M., 2017. Critical assessment of chitosan as coagulant to remove cyanobacteria. Harmful Algae 66: 1–12.

Lürling, M., Kang, L., Mucci, M., van Oosterhout, F., Noyma, N.P., Miranda, M., Huszar, V., Waajen, G., Manzi, M. 2020. Coagulation and precipitation of cyanobacterial blooms. Ecological Engineering 2020; 158: 106032.

Miranda, M., Noyma, N., Pacheco, F.S., de Magalhães, L., Pinto, E., Santos, S., Soares, M.F.A., Huszar, V.L., Lürling, M., Marinho, M.M., 2017. The efficiency of combined coagulant and ballast to remove harmful cyanobacterial blooms in a tropical shallow system. Harmful Algae 65: 27–39.

Noyma, N.P., de Magalhães, L., Furtado, L.L., Mucci, M., van Oosterhout, F., Huszar, V.L.M., Marinho,M.M., Lürling, M., 2016. Controlling cyanobacterial blooms through effective flocculation and sedimentation with combined use of flocculants and phosphorus adsorbing natural soil and modified clay. Wat. Res. 97: 26-38.

Noyma, N.P, de Magalhães, L., Miranda, M., Mucci, M., van Oosterhout, F., Huszar, V.L.M., Marinho, M.M., Lima, E.R.A., Lürling, M., 2017. Coagulant plus ballast technique provides a rapid mitigation of cyanobacterial nuisance. PLoS ONE 12(6): e0178976.

COMMENT

Page 8, line 161-163: The dose design of PAC, Chitosan, and MOS was different, would the difference of coagulation effect be affected by the dosage range of different coagulants? For example, the dose of 8 mg/L chitosan as shown in Fig. 3 has not been proved to reach the optimal dose with stable removal efficiency, while the span of MOS dose is larger with 20 mg/L has a relatively high efficiency and 30 mg/L has reached the optimal dose.

RESPONSE

We appreciate the constructive comment forwarded by the reviewer. We have chosen the dose of PAC and chitosan based on the results of previous studies. In those studies , strong reduction in PSII efficiency was found at a PAC dose of 4 mg/L or higher (Noyma et al., 2016), or 8 mg/L or higher (e.g. Miranda et al., 2017). Moreover, chitosan may also impair cell membrane integrity at higher doses (Mucci et al., 2017) and therefore, we did not include those higher doses in the tests performed here. Moreover, we hypothesized from the previous studies with freshwaters that a 2 mg Al/L and a chitosan dose of 2 mg/L would be effective. As regards MOS, we, however, learned that different studies used different, higher doses. Therefore, we did a two round dose estimation for MOS to estimate the effective dose . In the first round, we took the average values from different studies and evaluated doses of 0, 10, 50, 100, and 140 mg/L for their efficiency of cyanobacterial removal and cell lysis. Based on the results, we subsequently narrowed down the doses to 0, 10, 20, 30, 40, and 50 mg/L. We have added some sentences to the discussion to clarify our choice of test doses. 

COMMENTS 

7. Section 2.4: It is not recommended to use question sentence as subtitle in the experimental methods section.

RESPONSE

We accepted the comment and corrected the subtitle of section 2.4

COMMENT

8. Page 11, line 233-234: What does the “drop” and “remain” of PSII indicate? It should be explained.

RESPONSE

We have accepted the comment and added explanations for the “drop” and “remain” of PSII

COMMENT

9. Page 12, line 253: It is recommended to list the process and results of data analysis in a table or supporting information, and the supporting information in this paper seems to have uploaded the wrong file.

RESPONSE

We have accepted the comments and attached the results of data analysis as a supplementary file

COMMENT

10. Page 13, line 288: The “Linear regression model” is recommended to provide in detail.

RESPONSE

Thank you for this comment. It made us have a critical look at the data and that clearly revealed the growth observed was not following the exponential model. Hence, we have reanalyzed the data by iterative non-linear regression analysis using a logistic growth model. This provided far better estimates for growth rates in TRW and WC, but not in RRW where after 14 days a decline in Chl-a occurred. We have revised the methods, the results and the discussion accordingly.

COMMENT

11. Page 14, line 297: “8Al/L”, check unit.

RESPONSE

We accepted the comment and added the unit “mg ” to Al/L

COMMENT

12. Page 15, line 330-331: “At higher concentrations of 50 mg/L MOS and above, there was a sign of cell lyses with a high reduction of PSII efficiency”. However, Fig. 3 showed that PSII efficiency had dropped when the MOS dose is 20 mg/L. Confusing.

RESPONSE

We appreciated the critical comment. It is true that at MOS concentrations above 10 mg/L PSII efficiency started to decline from the initial value. However, compared to the dosage of 50 mg/L it was still considered marginal. We have rephrased the manuscript and also omitted the speculation on cell lysis as this was not measured. We also think that line 330–331 is confusing to readers. Therefore, we took out that sentence from the manuscript.

COMMENT

13. Page 16, line 345: Section 4.2, check subtitle.

RESPONSE

We accepted the comment and corrected the subtitle of section 4.2, page 354

COMMENT

14. Fig. 2 is not clear, and there is no directional significance of the actual results. What is the author’s intention to put Fig. 2 ?

RESPONSE

Thank you for your comments. Fig.2a was added with the intention of explaining the conceptual framework of the floc and sink assay. Fig. 2b shows the process of the floc and sink experiment and Fig.2c shows the level of treatment of effective coagulants. In figure 2c the right side picture shows the reservoir water before treatment, while the left side picture shows the reservoir water after treatment.

COMMENT

15. Fig. 3, “Moringa” in Fig.3B should be consistent with “MOS” in the text.

RESPONSE

We accepted the comment and adjusted Fig. 3B accordingly 

16. Fig.4, why not put MOS results in the same figure with PAC and chitosan? Using the drawing method of Fig. 3 to distinguish the top and bottom is a better choice.

RESPONSE

We accept the comment and adjust Fig. 4 accordingly

COMMENT

17. Fig.6 is not clear.

RESPONSE

Thank you for the comment Fig. 6 shows the growth rate of different cyanobacterial species in different media during the 26 days experimental period. We have modified the graph using similar y-axis in each panel and we have included the non-linear regression outputs (logistic growth model).

COMMENT

18. References: It would be better to quote more recent references such as in the last three years.

RESPONSE

Thank you for your comment. Some old references were replaced with more recent ones

Reviewer #2

COMMENT

Introduction: the authors give more information and description to support their assumption: water turbidity functions as ballast.

RESPONSE

Thank you for your comment. We have accepted the comment and added information regarding turbidity as a ballast on page 5, lines 88-91

COMMENT

The chemical composition of Moringa oleifera seed may change between those collected from different locations. This information is helpful to understand its efficiency.

RESPONSE

We have accepted the comment and included information on the chemical composition of Moringa seed available in Ethiopia in the “Materials and methods” part .

COMMENT

3) In Fig.3 A &C Chla concentration was higher in the high treatments than in the control for bottom and surface?

RESPONSE

Thank you for your comment Chl-a concentration was higher only in the bottom. The addition of the coagulants caused settling of the phytoplankton and suspended solids leading to higher Chl-a in the bottom samples, but lower in the top compared to the controls. 

COMMENT

4) In Fig.5，treated with MOS，PSII efficiency decreased at the surface, but increased for cyanobacteria at the bottom. Why?

RESPONSE 

We appreciate your keen observation and question. The drop in PSII efficiency is to slightly above 0.4. In those two treatments, comparatively most cyanobacteria cells, approximately 96%, were found in the bottom with coagulants, which creates more bright conditions in the top. A general response of phytoplankton, including cyanobacteria, to rather rapid increased light intensity is a reduction of PSII efficiency (Deblois et al., 2013). In the bottom, there are no differences in PSII efficiency. Alternatively, the cyanobacteria remaining in the top were not precipitated with ballast weighed down flocs and thus potentially exposed to relatively more binding sites of Moringa (and chitosan) causing some damage.

Deblois CP, Marchand A, Juneau P (2013) Comparison of Photoacclimation in Twelve Freshwater Photoautotrophs (Chlorophyte, Bacillaryophyte, Cryptophyte and Cyanophyte) Isolated from a Natural Community. PLoS ONE 8(3): e57139. doi:10.1371/journal.pone.0057139 

COMMENT

5) As turbidity is assumed to be ballast, it is better to give more physical and chemical information of the turbidity. Know which kinds of turbidity function better.

RESPONSE

We accepted the comment and provided some physicochemical features of turbidity 

COMMENT

6) In Fig.3, Fig.4, Fig.7, adding error bars.

RESPONSE

Thank you for your comment, when we revised the documents we removed Fig 7. Moreover, estimation of the dose of coagulants shown in Fig. 3, and measurement of the turbidity shown in Fig 4, were based on only single trial results for each dose; Therefore, we were unable to add error bars to Fig. 3. and Fig. 4

COMMENT

7) Redrawing Fig.6 to be clear for readers.

RESPONSE

We have accepted the comment and have redrawn Fig. 6.

COMMENT

8) The English needs correction. For example, Line 86: future =feature? Line 88-89: “The availability of local soils with natural P binding capacities was. For

instance, considered…” =” For instance, the availability of local soils with natural P binding capacities was considered… ”.

RESPONSE

Thank you for your comments. The English of Line 86 and Lines 88-89 was corrected. We have also made proof reading and correction of the manuscript accordingly.

---

## [Decision Letter · Decision Letter 1]

15 Mar 2021

PONE-D-20-31065R1

Removal of cyanobacteria from a water supply reservoir by sedimentation using flocculants and suspended solids as ballast: Case of Legedadi Reservoir (Ethiopia)

PLOS ONE

Dear Dr. Habtemariam,

Thank you for submitting your manuscript to PLOS ONE. After careful consideration, we feel that it has merit but does not fully meet PLOS ONE’s publication criteria as it currently stands. Therefore, we invite you to submit a revised version of the manuscript that addresses the minor points raised by one of the two reviewers.

We look forward to receiving your revised manuscript.

Kind regards,

Jean-François Humbert

Academic Editor

PLOS ONE

Journal Requirements:

Reviewers' comments:

Reviewer's Responses to Questions

**Comments to the Author**

1. If the authors have adequately addressed your comments raised in a previous round of review and you feel that this manuscript is now acceptable for publication, you may indicate that here to bypass the “Comments to the Author” section, enter your conflict of interest statement in the “Confidential to Editor” section, and submit your "Accept" recommendation.

Reviewer #1: All comments have been addressed

Reviewer #2: All comments have been addressed

2. Is the manuscript technically sound, and do the data support the conclusions?

Reviewer #1: Yes

Reviewer #2: Yes

3. Has the statistical analysis been performed appropriately and rigorously? 

Reviewer #1: Yes

Reviewer #2: Yes

4. Have the authors made all data underlying the findings in their manuscript fully available?

Reviewer #1: Yes

Reviewer #2: Yes

5. Is the manuscript presented in an intelligible fashion and written in standard English?

Reviewer #1: Yes

Reviewer #2: Yes

6. Review Comments to the Author

Reviewer #1: There are still some problems to be corrected. It is not limited to the problems listed below. Please read the full text carefully and revise the details.

1) Page 4, line 86-87: Grammar check: “it’s” ?

2) Page 6, line 122: 276 µg/L what? Check your parentheses use.

3) Page 6, line 123-124: “The reservoir has a mean total suspended solids (TSS) concentration of 390 mg/L and total suspended solids” ? Check the statement expression and the use of parentheses.

4) Page 6, line 125-126: Sentence is unreasonable, lack of punctuation.

5) Page 6, line 128: Unpublished data is not recommended for citation.

6) Page 7, line 130-133: According to what? “moisture content of 6.1 ± 0.2, oil content of 41.4 ± 1.6,” and “protein content 42.6 ± 1.4,” ? Check your expression.

7) Fig.1 is not clear.

8) Fig. 2a, there are no relevant instructions in the text, and we still feel that Figure 2 has no actual data to point to value.

9) Fig.6 is not clear, the legend in the picture is invisible, and error bars are not clear.

Reviewer #2: I checked the revision of he manusscript, the authors have corrected and well responded to my questions.

7. PLOS authors have the option to publish the peer review history of their article (what does this mean?). If published, this will include your full peer review and any attached files.

Reviewer #1: No

Reviewer #2: No

---

## [Author Response · Author response to Decision Letter 1]

20 Mar 2021

Responses to Reviewer 1

COMMENT 

There are still some problems to be corrected. It is not limited to the problems listed below. Please read the full text carefully and revise the details. 

RESPONSE

Thank you for your comment. While we were addressing the comments we have made revision of the entire manuscript regarding grammar and punctuation, For easy identification of track changes in the manuscript, we red-highlighted in bold all changes made 

COMMENT 

 Page 4, line 86-87: Grammar check: “it’s” ?

RESPONSE

Thank you for your comment. We have corrected the grammar

COMMENT

2) Page 6, line 122: 276 µg/L what? Check your parentheses use.

RESPONSE

Thank you for your comment. We have rephrased the sentence and corrected the use of parentheses 

COMMENT

3) Page 6, line 123-124: “The reservoir has a mean total suspended solids (TSS) concentration of 390 mg/L and total suspended solids” ? Check the statement expression and the use of parentheses.

RESPONSE

Thank you for your comment we have rephrased the sentence and corrected the use of parentheses

COMMENT

4) Page 6, line 125-126: Sentence is unreasonable, lack of punctuation.

RESPONSE

Thank you for your comment we have added a punctuation mark in the sentence

COMMENT

5) Page 6, line 128: Unpublished data is not recommended for citation.

RESPONSE

Thank you for your comment. Currently, we have published the manuscript, therefore, this citation is published data.

COMMENT

6) Page 7, line 130-133: According to what? “moisture content of 6.1 ± 0.2, oil content of 41.4 ± 1.6,” and “protein content 42.6 ± 1.4,” ? Check your expression.

RESPONSE

Thank you for your comment we have rephrased the sentence

COMMENT

7) Fig.1 is not clear.

RESPONSE

Thank you for your comment. Fig 1 is the description of the study area. We believed that it is important to include a visual map for the reader for a better understanding of the study. 

COMMENT

8) Fig. 2a, there are no relevant instructions in the text, and we still feel that Figure 2 has no actual data to point to value.

RESPONSE

Thank you for your comment. we have take-out figure 2 from the manuscript and put it to supplementary data

COMMENT

9) Fig.6 is not clear, the legend in the picture is invisible, and error bars are not clear.

RESPONSE

Thank you for your comment we have made all correction to Fig.6

---

## [Editor Report · Decision Letter 2]

24 Mar 2021

Removal of cyanobacteria from a water supply reservoir by sedimentation using flocculants and suspended solids as ballast: Case of Legedadi Reservoir (Ethiopia)

PONE-D-20-31065R2

Dear Dr. Habtemariam,

We’re pleased to inform you that your manuscript has been judged scientifically suitable for publication and will be formally accepted for publication once it meets all outstanding technical requirements.

Kind regards,

Jean-François Humbert

Academic Editor

PLOS ONE
---

## [Editor Report · Acceptance letter]

26 Mar 2021

PONE-D-20-31065R2 

Removal of cyanobacteria from a water supply reservoir by sedimentation using flocculants and suspended solids as ballast: Case of Legedadi Reservoir (Ethiopia) 

Dear Dr. Habtemariam:

I'm pleased to inform you that your manuscript has been deemed suitable for publication in PLOS ONE. Congratulations! Your manuscript is now with our production department. 

Kind regards, 

on behalf of

Dr Jean-François Humbert 

Academic Editor

PLOS ONE